# Identification and Functional Analysis of Two UGT84 Glycosyltransferases in Flavonoid Biosynthesis of *Carthamus tinctorius*

**DOI:** 10.3390/plants14193112

**Published:** 2025-10-09

**Authors:** Chaoxiang Ren, Jinxin Guo, Siyu Liu, Bin Xian, Yuhang Li, Changyan Yang, Cheng Peng, Jin Pei, Jiang Chen

**Affiliations:** 1State Key Laboratory of Southwestern Chinese Medicine Resources, Chengdu University of Traditional Chinese Medicine, Chengdu 611137, China; chaoxiangren@cdutcm.edu.cn (C.R.); 18982386850@163.com (J.G.); liusiyu0613@163.com (S.L.); xianbin@stu.cdutcm.edu.cn (B.X.); 15386562833@163.com (Y.L.); 17383100698@163.com (C.Y.); pengchengchengdu@126.com (C.P.); 2College of Pharmacy, Chengdu University of Traditional Chinese Medicine, Chengdu 611137, China

**Keywords:** *Carthamus tinctorius* L., glycosyltransferase, UGT84 subfamily, flavonoid biosynthesis, functional identification

## Abstract

Safflower (*Carthamus tinctorius* L.) is a multipurpose economic crop. Flavonoid glycosides are its key bioactive constituents, and several glycosyltransferases involved in their biosynthesis have been identified. The glycosyltransferase 84 subfamily represents a specialized branch with diverse functions, involved not only in catalyzing flavonoid glycosylation but also in the biosynthesis of auxins, tannins, and other compounds. However, this subfamily remains poorly characterized in safflower. In this study, two UGT84 subfamily genes, *UGT84A28* and *UGT84B3*, were screened based on expression patterns and phylogenetic evolution analysis. Recombinant proteins were induced and purified using prokaryotic expression systems. Functional characterization was subsequently conducted through enzymatic assays in vitro and transient expression in tobacco leaves. Molecular docking was employed to investigate the binding modes of UGTs with UDP-glucose. The results indicated that both UGTs demonstrated glycosylation activity at the flavonoid 7-OH position. Notably, when luteolin was employed as the aglycone, both enzymes also exhibited 3′-*O*-glycosylation activity. Combined with amino acid sequence alignment, we propose that residues A351/T343 and G263/F254, which affect spatial conformation and hydrogen bonding ability, may be one of the reasons for the functional differences between these two enzymes. These findings provide new insights into the catalytic diversity of glycosyltransferases.

## 1. Introduction

Safflower (*Carthamus tinctorius* L.) is an annual herbaceous plant belonging to the Asteraceae family. In traditional Chinese medicine, its flowers have long been recognized for their effects in promoting blood circulation, alleviating pain, and dispersing blood stasis. Modern pharmacological studies have further shown that safflower flowers exhibit multiple pharmacological activities, such as cardiovascular protection [1], anti-inflammatory effects [2], and immunomodulatory functions [3]. Beyond its medicinal value, safflower is also an important multipurpose economic crop. Its seeds can be processed into edible oil, while its flowers contain yellow and red pigments with applications in industry. Previous reports have revealed that flavonoids are the main components that determine the colors of safflower flowers [4].

To date, over 200 compounds have been isolated from safflower, among which flavonoids are recognized as the main chemical constituents and the principal active compounds responsible for its therapeutic effects. These flavonoids are mostly present in glycoside forms and can be categorized into special and common flavonoids [5]. Special flavonoids such as hydroxysafflor yellow A (HSYA), dehydrated safflor yellow B, and safflomin are *C*-glycosides with a chalcone backbone that are uniquely present in safflower. Common flavonoids include flavones, flavonols, dihydroflavones, and their derivatives, which are primarily present as *O*-glycosides and exhibit various biological activities.

The biosynthetic pathways of flavonoid compounds have been extensively studied [6], and an increasing number of genes involved in their biosynthesis pathways have been cloned and identified [7,8,9,10]. Glycosylation, typically the final step in the biosynthesis of natural products, is catalyzed by glycosyltransferases and plays a critical role in flavonoid glycoside biosynthesis. In plants, glycosylation reactions are primarily mediated by UDP-glycosyltransferases (UGTs), which transfer glycosyl groups from donors to acceptor aglycones to form glycosides [11]. The UGT family is the largest glycosyltransferase gene family in plants [12], catalyzing the majority of glycosylation reactions of natural products. UGTs are also involved in secondary metabolism [13], plant hormone regulation [14], and stress responses. The UGT superfamily can be divided into multiple families and subfamilies, which are further grouped based on phylogenetic relationships. Based on the modification sites in flavonoid glycosylation, most plant flavonoid UGTs are classified as 3-*O*-GT, 5-*O*-GT, 7-*O*-GT, 2′-*O*-GT, 3′-*O*-GT, 4′-*O*-GT, *C*-GT, and glycoside glucosyltransferases (GGTs), the latter being capable of catalyzing disaccharide or even polysaccharide formation.

The UGT84 subfamily represents a distinct category, with reported members exhibiting diverse catalytic functions. They can catalyze the formation of various flavonoid glycosides, including 7-*O*-glycosides, 4′-*O*-glycosides, and even *C*-glycosides, from multiple flavonoid aglycones [15,16]. They are also capable of synthesizing glucose esters [17]. For instance, galloylglucose esters produced from gallic acid (GA) and UDP-glucose are precursors of galloylated compounds, which are widely used in the food industry, biobased surfactants, cosmetics, and pharmaceuticals [18]. In addition, some UGT84 subfamily genes are involved in the synthesis of indole-3-acetic acid (IAA), thereby indirectly regulating plant growth and development [19]. Recent studies further reported that UGT84B1 participates in the biosynthesis of hydrolyzable tannins, forming a WRKY-DPB-UGT84B1 module that enhances drought tolerance [20].

Progress has also been made in identifying glycosyltransferases from safflower, such as UGT73AE1 [21], UGT71E5 [22], CtUGT3 [23], UGT95A2 [24], UGT88B2 [25], CtCGT [26], contributing to the understanding of flavonoid biosynthesis. However, research on the UGT84 subfamily in safflower remains limited, particularly regarding their roles in the biosynthesis of important bioactive compounds like flavonoids. In this study, we identified two genes belonging to the UGT84 subfamily in safflower through expression level and phylogenetic evolution analysis. Functional characterization demonstrated that both enzymes primarily catalyze flavonoid 7-OH glycosylation, while also synthesizing 4′-*O*-glycosides and even 3′-*O*-glycosides from specific flavonoid aglycones. These findings not only provide new insights into the biosynthesis of flavonoid compounds in safflower but also expand our understanding of the functional diversity of UGT84 subfamily genes.

## 2. Results

### 2.1. Mining and Bioinformatics Analysis of Candidate Genes

Based on transcriptome data from safflower flowers of different colors and from different tissues [4,10], two differentially expressed genes with high FPKM (fragments per kilobase of transcript per million mapped reads) values were screened, HH_023688 and HH_023694. HH_023688 was upregulated in deep red lines, and downregulated in white flowers. In contrast, HH_023694 was downregulated in deep red lines. Moreover, HH_023688 was specifically expressed in flowers and significantly upregulated at the fully bloomed stage. Similarly, HH_023694 was markedly upregulated in flowers, especially at the fully bloomed stage. Notably, HH_023694 was also upregulated by MeJA (Figure 1). The two transcripts encode 503 and 470 amino acids, with predicted molecular weights of approximately 55.9 and 52.1 kDa, and theoretical isoelectric points of 5.71 and 5.58, respectively (Appendix A).

Functional annotation, sequence alignment, and phylogenetic analysis revealed that HH_023688 and HH_023694 were named as UGT84A28 and UGT84B3, respectively. HH_023688 showed a closer genetic relationship with UGT84A23 and GtUF6CGT1, while HH_023694 was more closely related to UGT84B1 (Figure 2), suggesting that these candidate UGTs might share functional similarities with known members of the UGT84 subfamily.

### 2.2. Molecular Cloning, Recombinant Protein Expression, and Purification of UGTs

The full-length coding sequences of the two candidates were cloned at 1512 and 1413 bp, respectively. Two genes were constructed into the pET32 vector individually. Recombinant UGT proteins were expressed in the *E. coli* BL21 (DE3) strain and purified using a His-tag. SDS-PAGE analysis showed protein bands consistent with the expected size, confirming that two recombinant UGT proteins were successfully expressed and purified (Figure 3). Western blot analysis further verified the presence of recombinant UGT proteins (Appendix A). The purified proteins were used for further analysis.

### 2.3. Validation of the Enzymatic Activity of UGTs In Vitro

Flavonoid aglycones were used as sugar acceptors to detect enzyme activities, including apigenin, naringenin, and luteolin. The glycosylation reactions were detected by UPLC-MS. The glycoside products were identified by commercial standards. Both UGT84A28 and UGT84B3 showed 7-OH glycosylation activity, when apigenin (Figure 4) or naringenin (Figure 5) were used as the aglycones. UGT84A28 exhibited stronger activities than UGT84B3, especially with apigenin as the sugar acceptor.

When luteolin was used as the aglycone, UGT84A28 exhibited broader catalytic specificity producing three distinct glycosides (Figure 6A), luteolin-7-*O*-glucoside (retention time 7.31 min), luteolin-4′-*O*-glucoside (retention time 8.07 min), and luteolin-3′-*O*-glucoside (retention time 8.27 min), compared with luteolin itself (retention time 9.75 min). In contrast, UGT84B3 catalyzed luteolin almost exclusively to luteolin-3′-*O*-glucoside (retention time 8.24 min) (Figure 6B).

### 2.4. Biochemical Properties of UGT84A28 and UGT84B3

To examine the biochemical properties of UGT84A28 and UGT84B3, the effects of temperature, reaction time, and pH were systematically optimized using apigenin as the sugar acceptor and UDP-glucose as the sugar donor. Both enzymes exhibited rapid activity within the first 10 min, and the reactions were nearly complete within 60 min. Optimal enzymatic activity was observed in the temperature range 37–50 °C. Both UGTs displayed pH sensitivity, with maximum activity at pH 8–9. Specifically, UGT84A28 showed the highest activity in Tris-HCl buffer at pH 9, whereas UGT84B3 performed best in Na_2_CO_3_–NaHCO_3_ buffer at pH 9 (Figure 7).

### 2.5. Validation of UGT84A28 and UGT84B3 Activity in Tobacco

Transient expression vectors pEAQ-UGT84A28 and pEAQ-UGT84B3 were constructed via Gateway cloning and introduced into *Agrobacterium tumefaciens* GV3101 for infiltration into tobacco leaves. After cultivation, the tobacco leaves were collected for metabolite extraction and analysis. Transient expression of either UGT84A28 or UGT84B3 elevated the accumulation of apigenin-7-*O*-glucoside in tobacco leaves, with UGT84A28 in particular resulting in a marked multi-fold elevation (Figure 8). The results confirm that UGT84A28 and UGT84B3 are functionally active in planta and capable of catalyzing 7-OH glycosylation.

### 2.6. Molecular Docking of UGTs

Three-dimensional protein models of UGT84A28 and UGT84B3 were generated by AlphaFold, and molecular docking between the UGTs and UDP-glucose was performed using AutoDock Vina. The lowest-energy binding conformations were selected for analysis. In the model of UGT84A28-UDP-glucose complex, 11 residues interacted with UDP-glucose, including ASN22 (N22), ARG26 (R26), ILE261 (I261), GLY263 (G263), ASP264 (D264), TRP347 (W347), SER348 (S348), GLN350 (Q350), ALA351 (A351), SER370 (S370), and GLU373 (E373) (Figure 9A). Several of these residues were located within the PSPG motif. Specifically, TRP347, SER348 and ALA351 interacted with phosphate oxygens, GLN350 and SER370 formed polar contacts with the C4′ hydroxyls group of glucose, and GLU373 interacted with both phosphate oxygens and the C6′ hydroxyl group of glucose. Additional stabilizing interactions outside of the PSPG motif included ILE261 (uracil ring stabilization), ARG26 (uracil–phosphates region stabilization and C6′ hydroxyls interaction), ASN22 and ASP264 (phosphate oxygen interactions), and GLY263 (ribose hydroxyls contacts).

In the UGT84B3-UDP-glucose complex, 8 residues were involved: GLN23 (Q23), ASN27 (N27), ARG31 (R31), THR284 (T284), TRP339 (W339), SER340 (S340), THR343 (T343), and SER362 (S362) (Figure 9B). Four residues in the UGT84B3 were located within the PSPG motif: THR343 stabilized the uracil ring, SER362 interacted with the phosphate oxygens, while TRP339 and SER340 formed polar contacts with the C3′ hydroxyl group of glucose. Additional residues in the UGT84B3 outside the PSPG motif also contributed to binding, including GLN23 (stabilizing the uracil–phosphate region), ASN27 and ARG31 (hydroxyl and ribose interaction), and THR284 (phosphate oxygen interactions).

## 3. Discussion

Flavonoids represent the principal bioactive ingredients of safflower, accumulating predominantly in flowers and largely determining the color of safflower. Most flavonoids in plants occur as glycosides, with glycosylation at the 3-OH or 7-OH positions being the most common modifications. Glycosyltransferases constitute a large and functionally diverse enzyme family, encompassing numerous members with a wide range of activities. Therefore, safflower contains multiple flavonoid glycosyltransferases with distinct functional specificities. Identifying functionally divergent UGTs in safflower not only improves our understanding of flavonoid accumulation patterns but also facilitates enzymatic resources for application of glycosyltransferases in synthetic biology.

Considerable progress has been made in characterizing the safflower UGT gene family. The first identified enzyme, UGT73AE1 [21], demonstrated broad substrate specificity, catalyzing the formation of N-, O-, and S-glycosides. Subsequently, UGT71E5 was characterized to catalyze the conversion of abundant O-glycosides into N-glycosides and to glycosylate four flavonoids (including quercetin, naringenin, genistein, and phloretin) to yield di- or multi-O-glucosides [22]. Using multi-omics approaches, 34 UGT genes from subfamilies UGT71, UGT72, UGT75, UGT79, and UGT83 were systematically screened and functionally characterized [27]. Earlier studies identified CtUGT3 [23], which catalyzes the synthesis of astragalin, and UGT95A2 [24], which produces flavonoid 3′-*O*-glycosides with high efficiency, and CtCGT, identified as a key gene involved in the biosynthesis of hydroxysafflor yellow A (HSYA), a characteristic bioactive compound in safflower [26]. Despite these advances, knowledge of the UGT84 subfamily in safflower remains limited.

Phylogenetic analysis is an important tool for classifying UGT family members into distinct subfamilies based on conserved sequence motifs, enabling the prediction of biological characteristics and potential roles in biological systems. In this study, two candidate *UGT* genes were screened based on expressing patterns. Both were highly expressed in flowers, especially at full bloom. Interestingly, their expression differed among safflower varieties with different flower colors. *UGT84A28* showed high expression levels in white flowers but minimal expression in deep red varieties, whereas *UGT84B3* was preferentially expressed in deep red varieties. Phylogenetic assigned both genes to the UGT84 subfamily, which has not previously been investigated in safflower. UGT84A28 showed a closer genetic relationship with UGT84A23 and GtUF6CGT1, while HH_023694 clustered closely with UGT84B1. These relationships suggest that the two candidate genes might share similar functional characteristics with known UGT84 subfamily members.

Enzymes in the UGT84 subfamily exhibit diverse catalytic activities involved in flavonoid biosynthesis, mainly including the following types. For example, UGT84A23 catalyzes the production of 7-*O*-glycosides from apigenin or genistein [28]. UGT84A49 and UGT84A119 display predominant glycosylation activities at the 7-OH and 4′-OH [16], while UGT84B1 produces quercetin-7-*O*-glucoside in engineered *E. coli* [29]. Differently, GtUF6CGT1 and UGT84A57 show *C*-glycosylation activity at the 6-position of flavones such as apigenin and luteolin [15,30]. Therefore, based on reported studies, three common flavonoid aglycones in safflower, apigenin, luteolin and naringenin, were selected as substrates to identify the functions of two candidate UGTs in 84-subfamily. When apigenin and naringenin were used as substrates, both UGT84A28 and UGT84B3 catalyzed 7-OH glycosylation, rather than 6-*C* glycosylation. Interestingly, when luteolin was used as the substrate, both UGT84A28 and UGT84B3 glycosylated the 3′-OH position. Notably, UGT84A28 additionally produced 7-*O*- and 4′-*O*-glycosides, highlighting the typical catalytic promiscuity of glycosyltransferases, while UGT84B3 generated the 3′-*O*-glycoside exclusively. The results report the previously uncharacterized ability of UGT84 subfamily to catalyze flavonoid 3′-OH glycosylation, thereby expanding current knowledge of glycosyltransferase functional diversity.

To further confirm the roles of UGT84A28 and UGT84B3 in the synthesis of flavonoid glycosides in plants, tobacco, a widely used system, was employed for transient expression analysis. The results demonstrated that transient expression of UGT84A28 and UGT84B3 in tobacco leaves significantly enhanced the production of apigenin 7-*O*-glucoside. However, no significant increase in glycoside production from luteolin was detected, particularly for the 3′-*O*-glycosides, likely due to the relatively low affinity and limited catalytic efficiency of these enzymes toward luteolin at the 3′-OH position. These observations were consistent with in vitro enzyme activity assays, which might explain the lack of substantial glycosylated product accumulation in plants. Collectively, transient expression in tobacco leaves further confirmed that UGT84A28 and UGT84B3 promote glycosylation in the biosynthesis of flavonoid compounds, specifically at the 7-OH position of apigenin.

Amino acid sequence alignment (Appendix A) and molecular docking revealed that the PSPG motif plays a crucial and highly conserved role in UDP-glucose binding, with one notable exception at residue A351 in UGT84A28, which corresponds to T343 in UGT84B3. As threonine residue is more capable of forming hydrogen bonds than alanine, this substitution might partly explain functional divergence between the two enzymes. Additionally, conserved residues near the N-terminus, including N22 and R26 in UGT84A28 and their counterparts N27 and R31 in UGT84B3, also contribute to interactions with UDP-glucose. Furthermore, molecular docking results indicated that I261, G263, and D264 in UGT84A28 stabilize the UDP-glucose structure. Notably, G263 in UGT84A28 corresponds to F254 in UGT84B3, and the structural difference might represent another factor underlying their functional divergent catalytic behaviors.

Overall, the UGT84 subfamily exhibits broad functional versatility. They not only catalyze the formation of various flavonoid glycosides, including 7-*O*-glycosides, 4′-*O*-glycosides, and even *C*-glycosides [15,16,28,30], but are also capable of synthesizing glucose esters [17] and indole-3-acetic acid (IAA) [19,31]. In this study, UGT84A28 and UGT84B3 primarily exhibited flavonoid 7-OH glycosylation activity with varying efficiencies. Notably, when luteolin was used as the substrate, both enzymes also produced the 3′-*O*-glycoside products. Specifically, UGT84B3 generated exclusively luteolin 3′-*O*-glycoside. This finding enhances understanding of the UGT84 subfamily. However, the molecular basis of functional differences in flavonoid glycosylation among UGT84 subfamily members require further investigation. Future studies should explore whether UGT84 subfamily in safflower participate in the biosynthesis of IAA, glycolipids, or other specialized metabolites.

## 4. Materials and Methods

### 4.1. Plant Materials

Four safflower cultivars with different flower colors, white (W), yellow (Y), light red (LR, normal type), and deep red (DR), were used in this study. Safflower plants were cultivated at the medicinal botanical garden, Chengdu University of Traditional Chinese Medicine. Seeds were sown in mid-October 2023, and flowering began in mid-May 2024. Roots, stems, leaves and petals were collected on the first, second, third and fourth days after anthesis. For each sample, tissues from at least 5 plants, with consistent genetic backgrounds and growth rates were pooled. A 100 μM solution of methyl jasmonate (MeJA) was sprayed onto healthy safflower flowers at the third day after anthesis. Flowers were then enclosed in transparent plastic bags to retain volatile phytohormones and enhance absorption of the elicitor solutions. After 6 h of treatment, the plastic bags were removed, and samples of flowers were collected. All samples were immediately frozen in liquid nitrogen and stored at −80 °C.

### 4.2. Screening and Bioinformatics Analysis of Candidate Genes

Based on transcriptome data from safflower (*Carthamus tinctorius* L.) across different tissues and colors [4,10], two differentially expressed *UGT* genes with high FPKM in flowers were screened. The physicochemical properties of the two UGT enzymes were predicted and analyzed using the Expasy ProtParam tool (https://web.expasy.org/protparam/ on 7 November 2024). Various UGT sequences were downloaded from NCBI on 15 March 2023, and a UGT phylogenetic tree was constructed using Geneious Prime (version 2024.0.7).

### 4.3. Molecular Cloning of Two Candidate UGT Genes and Expression of Recombination Proteins

Total RNA was extracted using TRIzol, and first-strand cDNA was synthesized from total RNA with a cDNA reverse transcription kit (TaKaRa, Kyoto, Japan). Full-length CDSs of the two candidate genes were amplified from cDNA using primers designed with Primer Premier 5 software (version 5.0) (Appendix A). PCR was performed under the following conditions: initial denaturation at 95 °C for 3 min; 34 cycles of 95 °C for 30 s, 58 °C for 10 s, and 72 °C for 1 min; followed by a final extension at 72 °C for 5 min. The PCR products were then cloned into pET32 vector. Specific primers were designed with Primer Premier 5 software (version 5.0) (Appendix A). Following sequence verification, the pET32 recombinant vector was transformed into *E. coli* TSsetta (DE3) chemically competent cells via heat shock. The cells were grown in 1 L of LB medium containing 50 μg/mL ampicillin and 29 μg/mL chloramphenicol at 37 °C until OD600 reached 0.4–0.5. IPTG was then added to a final concentration of 0.12 mM to induce expression at 16 °C and 200 rpm for 16 h. Cells were harvested by centrifugation (6517 g, 15 min, 4 °C) and the supernatant was discarded. The cell pellet was resuspended in 30 mL of lysis buffer (25 mM Tris-Base, pH 8, 500 mM NaCl, 5 mM imidazole) and lysed by sonication on ice for 20 min. Cell debris was removed by centrifugation (12,516 g, 45 min, 4 °C). One milliliter of Ni-NTA Beads 6 Fast Flow was added to the supernatant, and the sample was incubated with the resin at 4 °C for 2 h to facilitate protein binding. The protein-resin mixture was then loaded onto a gravity column. After discarding the flow-through, the column was washed with approximately 30 mL of washing buffer (25 mM Tris-Base, pH 8, 100 mM NaCl, 20 mM imidazole). The tagged protein was eluted with approximately 17 mL of elution buffer (25 mM Tris-Base, pH 8, 100 mM NaCl, 250 mM imidazole). The recombinant proteins were concentrated using a 30 kDa ultrafiltration tube, supplemented with 10% (*v*/*v*) glycerol, fast-frozen in liquid nitrogen, and stored at −80 °C. SDS-PAGE was used to determine protein expressions.

To validate the expression of recombinant UGT proteins, bacterial lysates were analyzed by Western blot. After IPTG induction, *E. coli* BL21(DE3) cells expressing His-tagged UGT proteins were harvested by centrifugation (4000× *g*, 10 min) and resuspended in 1× SDS-PAGE loading buffer (62.5 mM Tris-HCl pH 6.8, 2% SDS, 10% glycerol, 0.01% bromophenol blue). Samples were boiled at 95 °C for 5 min and separated on a 12% SDS-PAGE gel, and transferred to a PVDF membrane (Millipore, Bedford, MA, USA) using a semi-dry transfer system at 15 V for 30 min. Membranes were blocked with 5% non-fat milk in TBST (20 mM Tris-HCl pH 7.6, 150 mM NaCl, 0.1% Tween-20) for 1 h at room temperature, followed by incubation with a primary anti-His antibody (1:5000 dilution, Abcam, Cambridge, UK) overnight at 4 °C. After washing with TBST (3 × 10 min), the membrane was incubated with a horseradish peroxidase (HRP)-conjugated secondary antibody (1:10,000 dilution, goat anti-mouse IgG, Abcam) for 1 h at room temperature. Protein bands were visualized using an ECL chemiluminescence kit (Thermo Scientific, Waltham, MA, USA) and imaged with a ChemiDoc XRS+ system (Bio-Rad, Hercules, CA, USA). Uninduced bacterial lysates and empty vector controls were included as negative controls to confirm specificity. 

### 4.4. Enzyme Activity Assays In Vitro

The enzymatic activity of UGTs was assayed using apigenin, naringenin, and luteolin as substrates for in vitro enzymatic reactions. Each 100 μL reaction system was configured with 2 mM aglycones, 4 mM UDP-Glc, 10 μM purified UGT protein, and 50 mM Tris–HCl buffer (pH 9). Reactions were conducted at 37 °C for 1 h and terminated by adding 300 μL pre-chilled methanol. After centrifugation, the supernatant was then analyzed by UPLC-Q-Exactive Orbitrap/MS high-resolution mass spectrometry. Chromatography system was carried out on an UltiMate 3000 UPLC, with a ZORBAX Eclipse Plus C18 column (150 mm × 3.0 mm, 1.8 μm) maintained at 25 °C. The mobile phase consisted of 0.1% formic acid in water (A) and acetonitrile (B), using the following gradient: 0–40 min, 5–95% B; 40–45 min, 95% B; 45–50 min, 95–5% B. The flow rate was 0.2 mL/min, and absorbance was detected at wavelengths of 254, 280, 300, and 360 nm.

### 4.5. Determination of the Biochemical Properties

The biochemical properties of UGT84A28 and UGT84B3 were evaluated using a 100 μL reaction mixture containing 2 mM apigenin as the sugar acceptor, 4 mM UDP-glucose as the sugar donor, and 10 μM purified protein. To determine the optimal reaction time for UGTs, the enzymatic reactions were performed for 1, 5, 10, 20, 30, 60, 120 and 180 min. For optimizing the reaction temperature, enzyme reactions were performed at various temperatures (4, 16, 25, 37, 50, 70 °C). To examine the effect of pH, enzymatic reactions were conducted in buffers with different pH ranges: 6.0–8.0 (50 mM Na_2_HPO_4_-NaH_2_PO_4_ buffer), 7.0–9.0 (50 mM Tris–HCl buffer), and 9.0–11.0 (50 mM Na_2_CO_3_–NaHCO_3_ buffer). All experiments were performed in triplicates. Reactions were terminated by ice-cold MeOH and centrifuged at 21,500× *g* for 15 min. The supernatants were analyzed using UPLC.

### 4.6. Enzyme Activity Validation in Nicotiana benthamiana

Two *UGT* genes were first cloned into the pDONR207 vector via the BP reaction utilizing Gateway BP Clonase II Enzyme Mix (Thermo Fisher, Waltham, MA, USA). Subsequently, the *UGT* genes were introduced into the pEAQ11 vector through the LR reaction. Primers were listed in Appendix A. Recombinant and empty pEAQ vector were transformed into *Agrobacterium tumefaciens* GV3101 cells. Briefly, 500 ng of plasmid DNA was added to 100 μL competent GV3101 cells. The mixture was gently mixed, incubated on ice for 10 min, quickly frozen in liquid nitrogen for 5 min, then incubated in a 37 °C water bath for 5 min, and finally placed on ice for another 5 min. After recovery in 900 μL of LB liquid medium at 28 °C with shaking at 200 rpm for 3 h, 100 μL culture was plated on LB agar plates containing 50 μg/mL kanamycin and 20 μg/mL rifampicin, and incubated at 28 °C for 3 days. The agroinfiltration process in *Nicotiana benthamiana* was carried out as follows. Bacteria cultures were initiated from fresh colonies or glycerol stocks in 10 mL of LB medium supplemented with selective antibiotics. After incubation, the cultures were centrifuged at 2000 rpm for 15 min and the supernatant was discarded. The cells were then resuspended in 15 mL agroinfiltration solution and centrifuged again at 2500 rpm for 15 min. The supernatant was removed, and the cells were resuspended in 10 mL of agroinfiltration solution containing 200 μM acetosyringone. After a 2 h incubation at room temperature with gentle shaking in the dark, the optical density (OD600) of each culture was measured and adjusted to 0.2 using the same agroinfiltration solution, in a final volume of 10 mL.

Tobacco plants, grown in a greenhouse for 4–6 weeks were used for bacterial infiltration. To facilitate stomatal opening, the leaves were sprayed with water one day before infiltration. On the day of infiltration, bacterial suspensions were aspirated into disposable syringes without needles and gently injected into the abaxial side of the leaves. For each experimental group, three tobacco plants were infiltrated with three leaves per plant. The tobacco plants were kept in the dark for 1 day post-infiltration and then returned to standard growth conditions for an additional 4 days. A substrate solution containing 1 mM apigenin and 2 mM UDP-glucose was then prepared and infiltrated into the pre-treated tobacco leaves. The leaves were maintained for one more day, after which they were harvested, freeze-dried using a lyophilizer, and ground into a fine powder.

For metabolite extraction, 100 mg of leaf powder was mixed with 2 mL of 70% methanol, vortexed for 5 min, and extracted in a 60 °C water bath with ultrasonic treatment for 30 min. The mixture was centrifuged at 5000 rpm for 15 min at room temperature, and the supernatants were collected for analysis. The extracts were analyzed using UPLC-Q-Exactive Orbitrap/MS high-resolution mass spectrometry under the same chromatographic conditions as those applied in the enzyme activity assays.

### 4.7. Molecular Docking

The three-dimensional structures of the UGTs were constructed using AlphaFold 3 (alphafoldserver.com) on 6 May 2025. Molecular docking of UDP-glucose with UGT84A28 and UGT84B3, respectively, was performed using AutoDock Vina (version 1.2.5) [32,33]. The docking results were visualized using PyMOL [34].

## 5. Conclusions

In summary, two glycosyltransferase genes from the UGT84 subfamily in safflower, namely UGT84A28 and UGT84B3, were selected based on expression level and phylogenetic evolution analysis. Using in vitro enzymatic assays and transient expression in tobacco leaves, both UGTs were confirmed to primarily facilitate flavonoid 7-OH glycosylation. Notably, when luteolin was used as the aglycone, both enzymes exhibited unusual 3′-*O*-glycosylation activity. Molecular dockings were employed to investigate the binding modes of UGT84A28 and UGT84B3 with UDP-glucose. Combined with amino acid sequences alignment, we propose that residues A351/T343 and G263/F254, which affect spatial conformation and hydrogen bonding ability, may be one of the reasons for the functional differences between these two enzymes. This study identifies two UGT genes from the UGT84 subfamily in safflower, not only enriching our understanding of flavonoid biosynthesis in this plant but also demonstrating a novel 3′-OH glycosylation activity within the UGT84 subfamily. This finding provides new insights into the catalytic diversity of glycosyltransferases and expands the known functional repertoire of the UGT84 subfamily.

## Figures and Tables

**Figure 1 plants-14-03112-f001:**
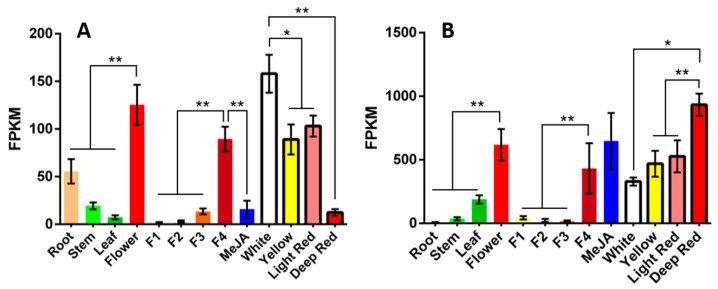
The expression patterns of two candidate *UGT* genes. (**A**) The expression pattern of *UGT84A28*; (**B**) The expression patterns of *UGT84B3*. *X* axis from left to right presented safflower root, stem, leaf, flower, the first day of flowering (F1), the second day of flowering (F2), the third day of flowering (F3), the fourth day of flowering (fully bloomed, F4), flower treated by MeJA, white flower lines, yellow flower lines, light red lines and deep flower lines, respectively. (* *p* < 0.05, ** *p* < 0.01).

**Figure 2 plants-14-03112-f002:**
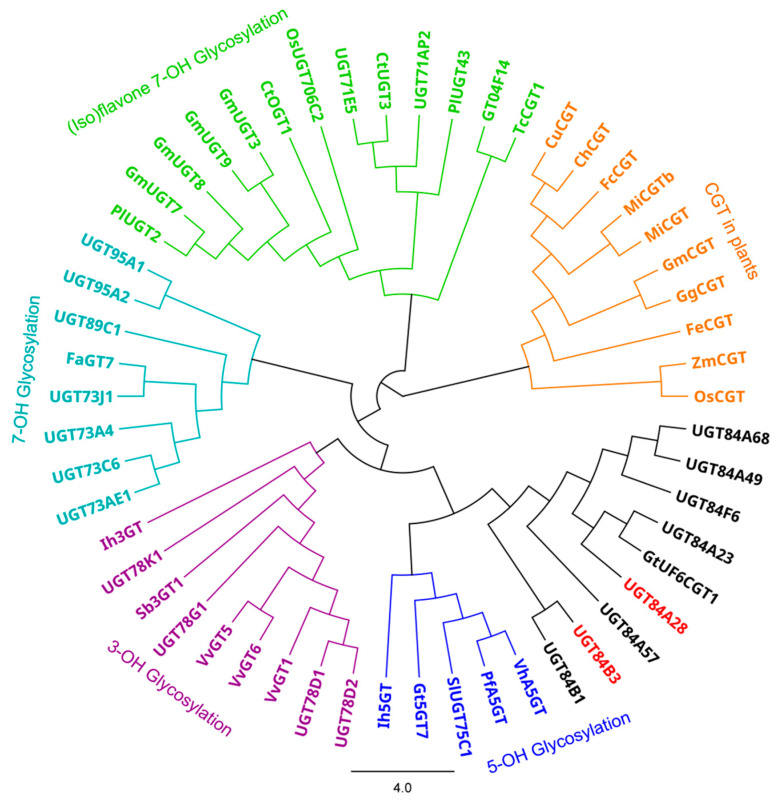
Phylogenetic analysis of UGTs. The phylogenetic tree was constructed from a Geneious Prime program (version 2024.0.7) using a neighbor-joining method (with 1000 bootstrap replications). Accession numbers and species names of UGTs used for the alignment are as follows. ChCGT (LC131335.1, *Citrus hanaju*), CuCGT (A0A224AKZ9, *Citrus unshiu*), CtUGT3 (WCF97139.1, *Carthamus tinctorius*), FaGT7 (Q2 V6J9, *Fragaria* × *ananassa*), FcCGT (A0A224AM54, *Citrus japonica*), FeCGT (AB909375.1, *Fagopyrum esculentum*), GgCGT (MH998596, *Glycyrrhiza glabra*), GmCGT (LC003312.1, *Glycine max*), GmUGT3 (AB904892.1, *Glycine max*), GmUGT7 (NM_001317558.2, *Glycine max*), GmUGT8 (NM_001292092.1, *Glycine max*), GmUGT9 (AB904896.1, *Glycine max*), GT04F14 (HQ219042.1, *Pueraria montana* var. *lobata*), Gt5GT7 (B2NID7, *Gentiana trifloral*), GtUF6CGT1 (A0A0B6VIJ5, *Gentiana trifloral*), MiCGT (XP_044491083, *Mangifera indica*), MiCGTb (KT989668.1, *Mangifera indica*), OSCGT (C3W7B0, *Oryza sativa*), OsUGT706C2 (DP000009.2, *Oryza sativa*), PfA5GT (Q9ZR27.1, *Perilla frutescens* var. *crispa*), PlUGT2 (A0A172J2D0, *Pueraria montana* var. *lobata*), PlUGT43 (A0A172J2G3, *Pueraria montana* var. *lobata*), Ih3GT (AB161175.1, *Iris hollandica*), Ih5GT (AB113664.1, *Iris hollandica*), Sb3GT1 (MK577650.1, *Scutellaria baicalensis*), SlUGT75C1 (NM_001361345.1, *Solanum lycopersicum*), TcCGT1 (PRJNA532685, *Trollius chinensis*), UGT71AP2 (8HOK_A, *Scutellaria baicalensis*), UGT71E5 (KX610759.1, *Carthamus tinctorius*), UGT73A4 (Q40286, *Manihot esculenta*), UGT73AE1 (KJ956788.1, *Carthamus tinctorius*), UGT73C6 (NP_181217, *Arabidopsis thaliana*), UGT73J1 (AY262063.1, *Allium cepa*), UGT78D1 (NM_102790.4, *Arabidopsis thaliana*), UGT78D2 (NM_121711.5, *Arabidopsis thaliana*), UGT78G1 (XM_003610115.4, *Medicago truncatula*), UGT78K1 (GU434274.1, *Glycine max*), UGT84A23 (ANN02875.1, *Punica granatum*), UGT84A49 (Q2V6K1, *Fragaria* × *ananassa*), UGT84A57 (BBI55602.1, *Eutrema japonicum*), UGT84A68 (UOL66843.1, *Ziziphus jujuba* var. *spinosa*), UGT84B1 (OAP11221.1, *Arabidopsis thaliana*), UGT84F6 (QDM38904.1, *Glycyrrhiza uralensis*), UGT89C1 (Q9LNE6, *Arabidopsis thaliana*), UGT95A1 (EU561020.1, *Hieracium pilosella*), VhA5GT (Q9ZR25.1, *Verbena hybrida*), VvGT6 (NM_001280903.1, *Vitis vinifera*), VvGT5 (AB499074.1, *Vitis vinifera*), VvGT1 (NP_001384786.1, *Vitis vinifera*), ZmCGT (NCVQ01000007.1, *Zea mays*).

**Figure 3 plants-14-03112-f003:**
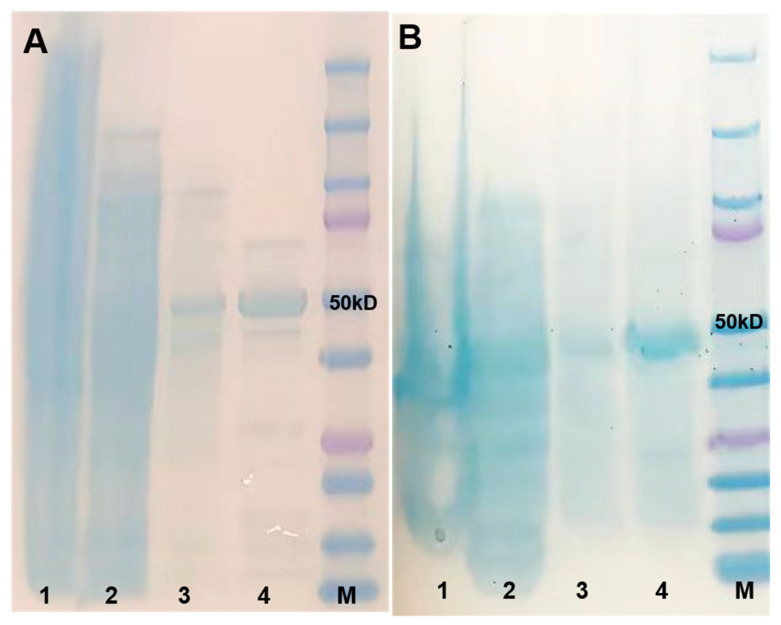
SDS-PAGE for two purified recombinant UGT proteins. (**A**) SDS-PAGE for UGT84A28 purification; (**B**) SDS-PAGE for UGT84B3 purification. Lane M, standard protein markers; Lane 1, disrupted cells; Lane 2, supernatant; Lane 3, the fraction eluted by wash buffer; Lane 4, the fraction eluted by elution buffer.

**Figure 4 plants-14-03112-f004:**
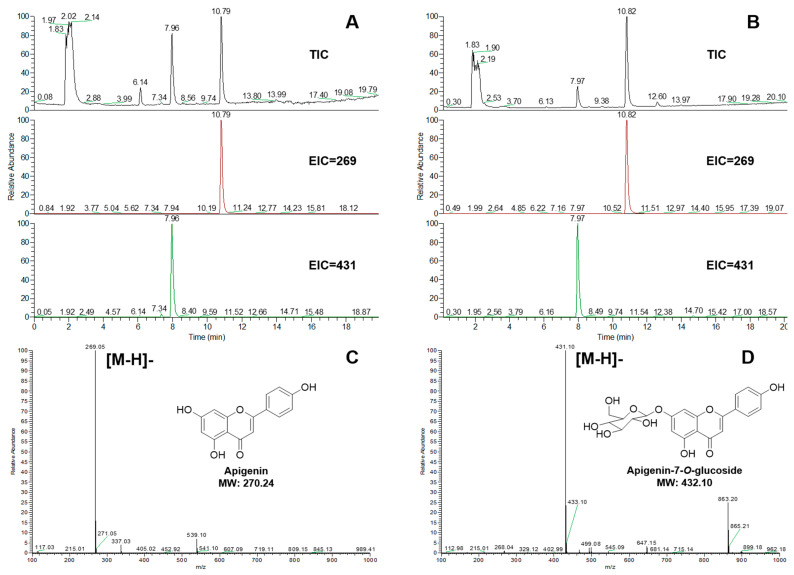
Mass spectrometry analysis of the reaction products of UGTs with apigenin. (**A**) TIC (total ion chromatogram) and EIC (extracted ion chromatogram) of the UGT84A28 reacted with apigenin (negative ion mode); (**B**) TIC and EIC of the UGT84B3 reacted with apigenin (negative ion mode); (**C**) Mass spectrometry of apigenin; (**D**) Mass spectrometry of apigenin-7-*O*-glucoside.

**Figure 5 plants-14-03112-f005:**
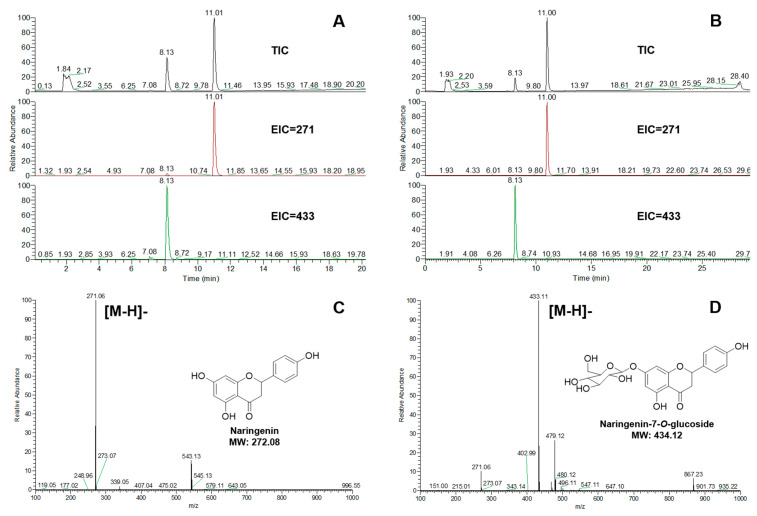
Mass spectrometry analysis of the reaction products of UGTs with naringenin. (**A**) TIC (total ion chromatogram) and EIC (extracted ion chromatogram) of the UGT84A28 reacted with naringenin (negative ion mode); (**B**) TIC and EIC of the UGT84B3 reacted with naringenin (negative ion mode); (**C**) Mass spectrometry of naringenin; (**D**) Mass spectrometry of naringenin-7-*O*-glucoside.

**Figure 6 plants-14-03112-f006:**
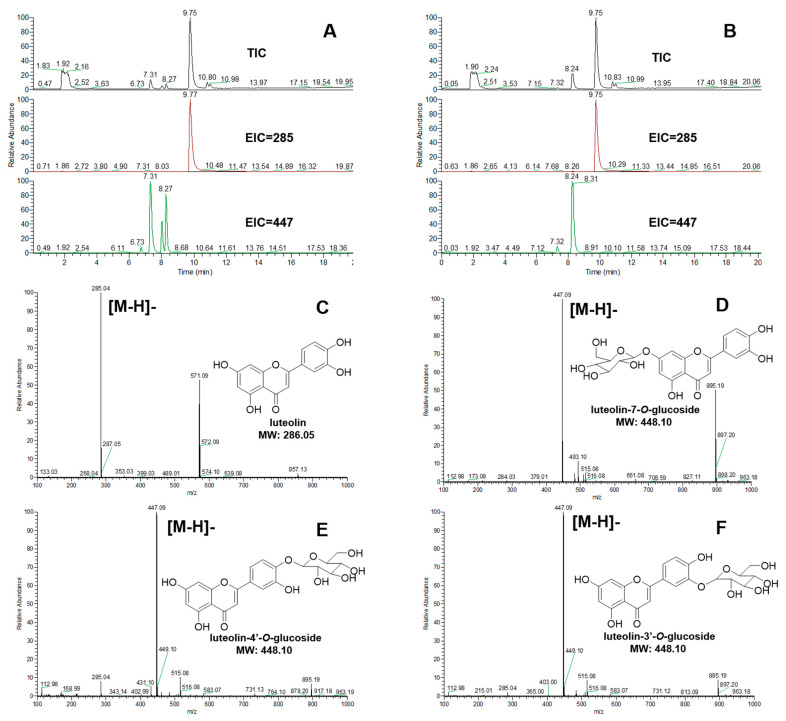
Mass spectrometry analysis of the reaction products of UGTs with luteolin. (**A**) TIC (total ion chromatogram) and EIC (extracted ion chromatogram) of the UGT84A28 reacted with luteolin (negative ion mode); (**B**) TIC and EIC of the UGT84B3 reacted with luteolin (negative ion mode); (**C**) Mass spectrometry of luteolin; (**D**) Mass spectrometry of luteolin-7-*O*-glucoside; (**E**) Mass spectrometry of luteolin-4′-*O*-glucoside; (**F**) Mass spectrometry of luteolin-3′-*O*-glucoside.

**Figure 7 plants-14-03112-f007:**
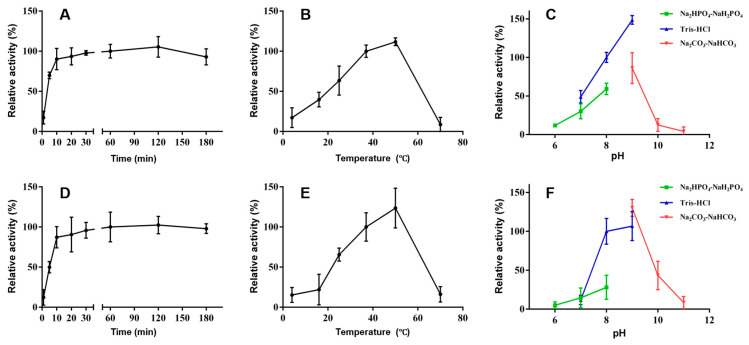
Biochemical properties of UGT84A28 and UGT84B3. (**A**–**C**) Effects of different reaction time, temperature, and pH on UGT84A28 activity; (**D**–**F**) Effects of different reaction time, temperature, and pH on UGT84B3 activity.

**Figure 8 plants-14-03112-f008:**
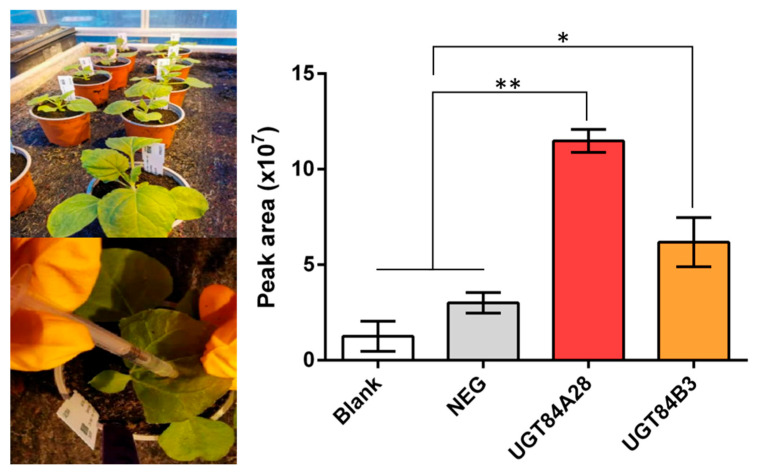
Apigenin-7-*O*-glucoside determination in tobacco leaves after transient expressing of UGTs. NEG represents the negative control group transferred to the empty vector. The differences are marked in the figure (* *p* < 0.05, ** *p* < 0.01).

**Figure 9 plants-14-03112-f009:**
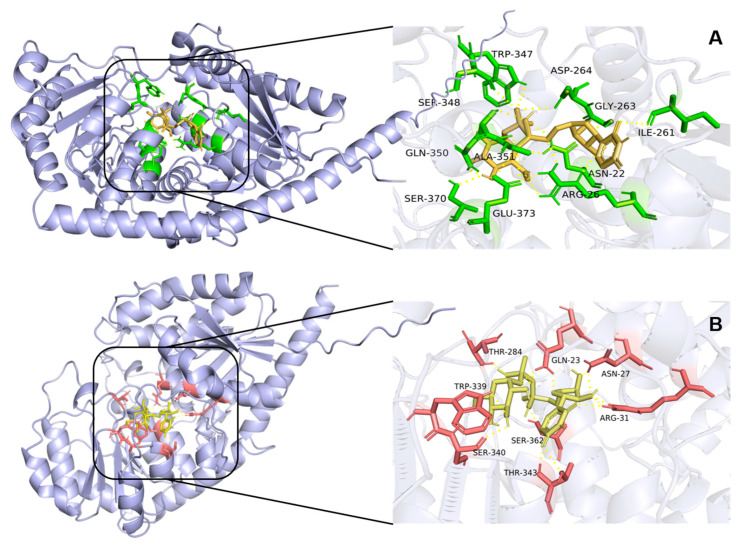
Models of UGTs binding with UDP-glucose. (**A**) Model of UGT84A28 binding with UDP-glucose. UDP-glucose was marked in yellow, and the residues that interacted with ligands were marked in green; (**B**) Model of UGT84B3 binding with UDP-glucose. UDP-glucose was marked in yellow, and the residues that interacted with ligands were marked in red.

## Data Availability

The original contributions presented in this study are included in the article/Appendix A. Further inquiries can be directed to the corresponding authors.

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
