# Peer review of "Identification and Functional Analysis of Two UGT84 Glycosyltransferases in Flavonoid Biosynthesis of Carthamus tinctorius"

_plants, 2025, doi:10.3390/plants14193112_

Round 1

Reviewer 1 Report

Comments and Suggestions for Authors

The manuscript “Identification and functional analysis of two UGT84 glycosyl-transferases in flavonoid biosynthesis of Carthamus tinctorius” by Ren et al, aimed to explore the unknown UGT84 glycosyltransferases. In this study, the authors identified two games belonging to UGT84 subfamily in safflower and validated their primary catalyze functions. 

This study is very important, it provides insights into functional diversity. But there are some minor concerns that should be address: 

1. Line 108: In Figure 1, the authors should add the statistical analysis, to show the significance of the expression patterns. And it is fully understood that the authors try to use different colors representing the types of tissues. For that purpose, please add the color legends, and put the “F1”, “F2”, “F3” as the full name. 

2. Line 120: In figure 2, please add the figure legend in the phylogenetic tree to show the distance. And if different colors represents the sub-groups, please clarify the details. 

3. Line 154: In figure 3, please add the negative control (empty plasmid with induction) and uninduced disrupted cells

4. Line 347: please briefly describe the physiological stage of materials when the author collected them, and the amount of the materials and related treatment before the frozen at -80. 

5. Line 356: please list the UGT sequences downloaded from NCBI in the supplementary material.

6. Line 355: did the authors use the default setting for the analysis? Otherwise, please give the description.

Comments on the Quality of English Language

Please proofread the whole manuscript and edit the grammar errors. 

Author Response

Comments 1: Line 108: In Figure 1, the authors should add the statistical analysis, to show the significance of the expression patterns. And it is fully understood that the authors try to use different colors representing the types of tissues. For that purpose, please add the color legends, and put the “F1”, “F2”, “F3” as the full name. 

Response 1: Thanks for you comment. We have added the statistical analysis and the color legends in Figure 1 in the revised manuscript.

Comments 2: Line 120: In figure 2, please add the figure legend in the phylogenetic tree to show the distance. And if different colors represents the sub-groups, please clarify the details. 

Response 2:  Thanks for you comment. We have revised this figure in the revised manuscript. 

Comments 3: Line 154: In figure 3, please add the negative control (empty plasmid with induction) and uninduced disrupted cells

Response 3: Thanks for you comment. We have supplied a western bolt analysis by anti-His antibody include the negative control (empty plasmid with induction) and uninduced disrupted cells. The image has listed in supplementary materials.

Comments 4: Line 347: please briefly describe the physiological stage of materials when the author collected them, and the amount of the materials and related treatment before the frozen at -80. 

Response 4: Thank you for pointing this out. We have provided detailed sample information in section 4.1 in the revised manuscript.

Comments 5: Line 356: please list the UGT sequences downloaded from NCBI in the supplementary material.

Response 5: Thanks for you comment.  In the phylogenetic analysis, it included 50 UGT sequences. If we list these 50 complete sequences, it will take up a lot of space, and we believe it is unnecessary. Therefore, we have listed the accession numbers for each UGT in the legend of Figure 2, which can be used to search for complete sequences and related information in NCBI.

Comments 6: Line 355: did the authors use the default setting for the analysis? Otherwise, please give the description.

Response 6: Thank you for pointing this out. The analysis performed by the Expasy ProtParam tool used default setting. Therefore, we didn't give the description. For phylogenetic analysis, most settings used default values, and special settings were described in Figure 2 legend.

Reviewer 2 Report

Comments and Suggestions for Authors

Manuscript titled “Identification and functional analysis of two UGT84 glycosyl-transferases in flavonoid biosynthesis of Carthamus tinctorius” reports a phylogenetic and functional analysis of safflower glycosyl transferases. The authors validated the activity of the enzymes under different pH and temperature, and determined their affinity to produce different glycosylation patterns. The work is interesting and relevant; there are some minor comments and suggestions for the authors:

  1. There is a repeated phrase in lines 48-49, please restructure them.
  2. Lines 104 and 111 mention the use of “MeJA”, however, this is not mentioned in materials and methods, please include it where appropriate.
  3. Line 181 states that “When luteolin was used as a glycoside…”, should this phrase be “When luteolin was used as a sugar acceptor” instead? Please check this phrase and confirm.
  4. In section 4.1, do you know the time from when the seeds were planted until flower collection? Including time of year when this was performed and growing conditions is also advised if this data is available.
  5. Lines 382 and 396 mention that “10 μM” of purified protein was used. Are these units correct or should they be in mass instead of molarity for the purified protein? Please check and confirm if this is correct.

Author Response

Comments 1: There is a repeated phrase in lines 48-49, please restructure them.

Response 1: Thank you for pointing this mistake out. We have revised this phrase in the revised manuscript.

Comments 2: Lines 104 and 111 mention the use of “MeJA”, however, this is not mentioned in materials and methods, please include it where appropriate.

Response 2: Thank you for pointing this out. We have modified section 4.1 to explain how MeJA was used in plant materials.

Comments 3: Line 181 states that “When luteolin was used as a glycoside…”, should this phrase be “When luteolin was used as a sugar acceptor” instead? Please check this phrase and confirm.

Response 3: Thank you for pointing this mistake out. We have revised this sentence in the revised manuscript.

Comments 4: In section 4.1, do you know the time from when the seeds were planted until flower collection? Including time of year when this was performed and growing conditions is also advised if this data is available.

Response 4: Thanks for your comment. We have added the time from when the seeds were planted until flower collection in the revised manuscript. However, due to the experimental materials being cultivated in the medicinal plant garden, the cultivation conditions are unavailable.

Comments 5: Lines 382 and 396 mention that “10 μM” of purified protein was used. Are these units correct or should they be in mass instead of molarity for the purified protein? Please check and confirm if this is correct.

Response 5: Thanks for your comment. Since different proteins have varying molecular weights, using mass as a unit does not accurately reflect enzymatic activity. Therefore, we consider molarity to be more appropriate and have retained molar units for the recombinant proteins.

Reviewer 3 Report

Comments and Suggestions for Authors

The manuscript ID: plantss-3863393 entitled ”Identification and functional analysis of two UGT84 glycosyl-2 transferases in flavonoid biosynthesis of Carthamus tinctorius’’ by Ren et al. identified two genes belonging to the UGT84 subfamily in safflower through expression level and phylogenetic  evolution analysis. The functions of these two glycosyltransferases demonstrating that they primarily catalyze flavonoid 7-OH glycosylation. Additionally, the authors showed that for certain flavonoid aglycones, they also synthesize 4′-O-glycosides and even 3′-O-glycosides. I consider that the idea is good, and the results obtained are well presented and valorized. However, the paper needs some clarifications and improvement before acceptance for publication. Below you can find to be considered the relevant points:

Major points

  • Abstract should reflect the results obtained in the paper and not a descriptive of approaches. The abstract should be rewritten.
  • Why the authors investigate the expression patterns of the two candidate UGT genes under MeJA? What about the others phytohormones?
  • At the experimental level, it is not clear what is the difference between TIC (total ion chromatogram) and EIC (extracted ion chromatogram) and what is the difference between EIC 271 and EIC433?
  • I didn’t understand the difference between figure 5 and 6 in relation with EIC values?
  • Why the authors choose to investigate the functional role of UGT84A28 and UGT84B3?
  • A western blot experiment is strongly needed to confirm the identity of the two proteins even with His-tag antibody.
  • The details of molecular cloning of the two UGT should be added in the subheading 4.3

Minor points

  • Line 97, please define the abbreviation ‘’ FPKM values’’.
  • The quality of figure 3 is very low and a new figure is needed with high quality.
  • Figure 7, in the X axis of A, B, D and E, include space (Time (min)….).
  • Figure 8, please detail the abbreviation ‘’NEG’’.
  • Line 294, please check the word ‘’qucetein-7-O-glucoside’’.
  • Please added the date (D, M and Y) of accession to the different bioinformatic programs.
  • Species should be written in italic such as Agrobacterium tumefaciens.
  • English editing is necessary for the improvement of the quality of the paper.
Comments on the Quality of English Language

English language should be improved .

Author Response

Comments 1: Abstract should reflect the results obtained in the paper and not a descriptive of approaches. The abstract should be rewritten.

Response 1: Thanks for your comment. We have made revisions to the abstract in the revised manuscript. Due to the requirements of the journal, the abstract needs to be written in three parts: research background, research methods, and research results, and is required to be within 200 words. Therefore, we are unable to make significant modifications.

Comments 2: Why the authors investigate the expression patterns of the two candidate UGT genes under MeJA? What about the others phytohormones?

Response 2: Thanks for your comment. In previous study (Chen, J.; Tang, X.; Ren, C.; Wei, B.; Wu, Y.; Wu, Q.; Pei, J. Full-length transcriptome sequences and the identification of putative genes for flavonoid biosynthesis in safflower. BMC Genomics. 2018, 19, 548.), we found that MeJA treatment of safflower significantly affects the changes in flavonoid components. MeJA is also a key signaling molecule that activates secondary metabolism in plants, particularly in the biosynthesis of flavonoids and terpenoids. Therefore, we investigated the expression patterns of the two candidate UGT genes under MeJA. We have provided a detailed description of MeJA treatment of safflower in section 4.1 in the revised manuscript.

Comments 3: At the experimental level, it is not clear what is the difference between TIC (total ion chromatogram) and EIC (extracted ion chromatogram) and what is the difference between EIC 271 and EIC433? I didn’t understand the difference between figure 5 and 6 in relation with EIC values?

Response 3: The TIC represents the total intensity of all ions detected by the mass spectrometer at each point in time. It provides a global overview of the sample composition but can be complex and contain signals from many unrelated compounds. The EIC is a chromatogram generated by extracting the signal intensity for a specific mass-to-charge ratio (m/z) or a very narrow m/z window. It is highly specific and allows us to monitor the abundance of a particular ion of interest, effectively filtering out background noise and signals from other compounds. In metabolic studies, EICs are crucial for tracking specific metabolites. EIC 271 represented the chromatogram tracks the ion with m/z 271. This value corresponds to the substrate naringenin [M-H]⁻. EIC 433 represented the chromatogram tracks the ion with m/z 433. This value corresponds to the product naringenin-7-O-glucoside [M-H]⁻. The difference of 162 Da (433 - 271) is consistent with the addition of a hexose sugar unit (e.g., glucose), which is the expected reaction catalyzed by the glycosyltransferases we studied. The substrate in Figure 6 was luteolin (MW=286, EIC 285 [M-H]⁻) .

Comments 4: Why the authors choose to investigate the functional role of UGT84A28 and UGT84B3?

Response 4: Thanks for you comments. We introduced the uniqueness of the UGT84 subfamily in the introduction section (line 70-80). They not only catalyze the formation of various flavonoid glycosides, including 7-O-glycosides, 4′-O-glycosides, and even C-glycosides, from multiple flavonoid aglycones, but are also capable of synthesizing glucose esters and indole-3-acetic acid (IAA). However, research on the UGT84 subfamily in safflower remains limited. We screened two genes belonging to the UGT84 subfamily in safflower through expression level and phylogenetic evolution analysis, and named two genes, UGT84A28 and UGT84B3, based on the commonly used naming principles for glycosyltransferases.

Comments 5: A western blot experiment is strongly needed to confirm the identity of the two proteins even with His-tag antibody.

Response 5: Thanks for you comments. We have added a western blot experiment in the revised manuscript. A western blot image was added in supplementary materials.

Comments 6: The details of molecular cloning of the two UGT should be added in the subheading 4.3

Response 6: Thank you for pointing this out. We have added the details of molecular cloning in section 4.3.

Comments 7: Line 97, please define the abbreviation ‘’ FPKM values’’.

Response 7: Thank you for pointing this out. We have added a description in the revised manuscript.

Comments 8: The quality of figure 3 is very low and a new figure is needed with high quality.

Response 8: Thank you for pointing this out. We have reuploaded a new figure 3 in the revised manuscript.

Comments 9: Figure 7, in the X axis of A, B, D and E, include space (Time (min)….).

Response 9: Thank you for pointing this out. We have revised this figure in the revised manuscript.

Comments 10: Figure 8, please detail the abbreviation ‘’NEG’’.

Response 10: Thank you for pointing this out. We have added a description in the revised manuscript.

Comments 11: Line 294, please check the word ‘’qucetein-7-O-glucoside’’.

Response 11: Thank you for pointing this out. We have revised this mistake in the revised manuscript.

Comments 12: Please added the date (D, M and Y) of accession to the different bioinformatic programs.

Response 12: Thank you for pointing this out. We have added a description in the revised manuscript.

Comments 13: Species should be written in italic such as Agrobacterium tumefaciens.

Response 13: Thank you for pointing this out. We have revised this mistake in the revised manuscript.

Comments 14: English editing is necessary for the improvement of the quality of the paper.

Response 14: Thanks for your comment. The manuscript has been thoroughly edited by a professional English editing service, We have carefully reviewed and incorporated all the suggested changes. The revisions have significantly improved the grammar, spelling, sentence structure, and overall fluency of the manuscript. We believe the language quality of the revised manuscript now meets the high standards of the journal.

Round 2

Reviewer 3 Report

Comments and Suggestions for Authors

The paper was clearly improved and authors responded to all the queries.